# A Study on the Impact of Built Environment Elements on Satisfaction with Residency Whilst Considering Spatial Heterogeneity

Qi Chen [1,2], Yibo Yan [1], Xu Zhang [1] and Jian Chen [2,3,*]

1   College of Civil Engineering, Henan University of Technology, Zhengzhou 450001, China
2   College of Traffic & Transportation, Chongqing Jiaotong University, Chongqing 400074, China
3   Jiangsu Province Collaborative Innovation Center of Modern Urban Traffic Technologies, Nanjing 211189, China
*   Correspondence: chenjian@cqjtu.edu.cn

**Abstract:** The built environment, as perceived and felt by human beings, can shape and affect residential satisfaction. From the perspective of municipal administrators, understanding the building environment and its relationship with people's residential satisfaction is crucial to improving people's living environment. This study examines the correlation between built environment elements and residential satisfaction using the consideration of spatial heterogeneity of such a correlation. Machine vision technology is introduced to quantify the design dimension of the built environment. The method of multiscale geographically weighted regression is used to evaluate the relationship between built environment and residential satisfaction and to analyze the spatial heterogeneity in the influencing effects. This empirical study draws on 399 collected samples from the residents of Zhengzhou, China. The results show that elements of the built environment, including street space design features, have a significant effect on people's residential satisfaction in Zhengzhou City. The factors of functional diversity and distance to the city center show spatial heterogeneity in influencing effects on residential satisfaction. The results of this study could help municipal managers to improve people's residential satisfaction in Zhengzhou City through the development of urban renewal policies.

**Keywords:** urban environment; residential satisfaction; street view image; spatial heterogeneity; multiscale geographically weighted regression

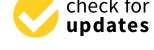



## 1. Introduction

Residential satisfaction has long been an important topic in urban studies. People's residential satisfaction not only affects their physical and mental health, but it is also a reflection of their sense of happiness [1,2]. Scholars generally agree that living in a comfortable, healthy, and convenient community can significantly increase people's residential satisfaction [3–5]. Additionally, the process of urban development, uncoordinated urban development, urban environmental pollution problems, and a deterioration in the quality of the living environment in urban areas will add considerable pressure to the urban development process [6,7]. Excessive urban living environment disparity has also led to an increase in dissatisfaction among the social underclasses. From the perspective of municipal managers, to reform the urban space, thus improving residents' living quality, it is necessary to identify the elements of the urban built environment that are closely related to residential satisfaction, so as to formulate targeted urban renewal strategies and plans.

According to Tobler's First Law of Geography, the closer matters are to each other, the stronger their correlation [8]. By extension, in the urban environment, there is also a certain spatial correlation between neighboring areas. Therefore, when exploring the influence of environmental factors, the spatial scale is important for the measurement of

environmental factors. It has been shown in the literature that some spatial heterogeneity exists in the environmental influences that affect people's residential satisfaction and physical activity [9–11]. Ignoring the scale differences of environmental factors in the urban environment can lead to inaccurate study results. Scale differences in the spatial heterogeneity of environmental influences should be fully considered when examining the link between environmental elements and residential satisfaction.

The purpose of this study is to investigate the influence of the built environment on residential satisfaction in Zhengzhou City and to establish the corresponding relationship between elements of the built environment and residential satisfaction. Thus, the residential satisfaction of Zhengzhou residents can be improved through targeted urban renewal strategies. The research contents are organized as follows: firstly, the extant research on the built environment and residential satisfaction are reviewed; secondly, the thought process, methodology, and indicators for this study are described; thirdly, empirical case data are analyzed; fourthly, the results and discussion are presented; lastly, the results from this study are concluded.

## 2. Literature Review

### 2.1. Impact Factors of Residential Satisfaction

In recent years, research topics on residential satisfaction have been focused on by scholars in the fields of sociology, geography, and urban planning. These studies can be divided into two categories: (1) using psychological research frameworks and methodologies to understand environmental issues [12,13] and (2) using statistical methodology and geography information data to explore the correlation between built environment elements and living satisfaction [14,15]. The former usually starts from psychological factors and responds to the psychological factors affecting people's satisfaction with living through questionnaires, while the latter establishes the link between the living environment and satisfaction with living through mathematical and statistical methods.

Considering psychological factors, Gamaldo analyzed the correlation between personal attributes and environmental perceptions of residential satisfaction through a survey of adults conducted in the Tampa area [12]. Additionally, Ciorici established a link between people's perceptions of the environment and residents' satisfaction by studying logistic regression models. It was confirmed that people's subjective satisfaction with the environment can significantly influence people's residential satisfaction [16,17]. Such studies provided a solid theoretical basis for urban planners, but they cannot provide refined improvement strategies as the correlation between built environment elements and residential satisfaction has not been established in previous studies.

To support urban renewal programs, scholars have paid more attention to the correlation between elements of the residential environment and residential satisfaction. With an in-depth understanding and exploration of the urban built environment, a 5D description framework has been gradually formed, namely, "Density, Diversity, Distance to transit, Destination accessibility, and Design [18]." However, due to a limitation in technical means and research methods, the 5D framework is described using geographic information data, which makes it difficult to express the design dimensions. For example, Olfindo studied the relationship between public transportation facilities and residential satisfaction by surveying residents living around stations and concluded that good transportation accessibility can improve residential satisfaction [14]. In recent years, research conducted on street view recognition, such as greenery and visual enclosure, based on machine vision has started to emerge in large numbers, which provides innovative ideas to complement the environmental impact factors in the design dimension [19,20]. The two factors of greenery and visual closure in the design dimension have been proven to help improve people's walking quality [21,22].

## 2.2. Residential Environment and Spatial Differences

At present, many empirical studies show that built environment elements as well as individual attribute variables have a significant influence on residential satisfaction. It has also been indicated that different people's perceptions of urban space may vary depending on the distribution of urban space. The researchers of such spatial differentiations hold that different communities may have differing attitudes towards the living environment due to the urban spatial distribution and that there is spatial heterogeneity between residential environment factors [11]. As such, neglecting spatial influence may result in inaccurate study results [23]. It is reasonable to think that there may also be some spatial differences in residential satisfaction due to the difference in perception of the urban built environment.

To tackle the spatial heterogeneity of people's perceptions, scholars introduced a geographically weighted regression (GWR) model to study the spatial distribution of the built environment. Since there might also be differing extents of spatial heterogeneity of the various influence factors, scholars further developed a multiscale geographically weighted regression (MGWR) model on the basis of the GWR model [24]. MGWR allows each variable to have a different bandwidth, which in turn produces more reliable estimates and provides a range of influence for different variables [25]. Compared with the GWR model, this analysis is more in line with the actual situation and has been used in a wide range of fields in recent years. The MGWR model has been widely applied to explore the spatial heterogeneity of a built environment's impact. For example, scholars used the MGWR model to explore whether spatial heterogeneity between the residential environment and household attributes exists in people's health status [9]. Additionally, scholars have used the MGWR model to examine the influence of the built environment and personal attributes on commuting behavior [26].

## 2.3. Literature Summary

Scholars have explored the relationship between the built environment and residential satisfaction from different perspectives, and research on the link between the built environment and residential satisfaction has been generally accepted by scholars [25–33] (Table 1). However, the following problems still remain in the field of residential satisfaction: (1) The research methods of residential satisfaction are mostly statistical methods, without considering the spatial heterogeneity of the influencing factors in terms of geography. It is necessary to examine the spatial heterogeneity of residential satisfaction influencing factors in cities with large differences in urban environments. (2) The current research on the built environment and residential satisfaction does not sufficiently consider environmental factors and especially lacks a measurement of the design dimension.

**Table 1.** Horizontal comparison of literature.

| Year | 2018 | 2018 | 2019 | 2019 | 2020 | 2020 | 2021 | 2021 | 2022 | 2022 | This study |
|---|---|---|---|---|---|---|---|---|---|---|---|
| Author | Feng [27] | ElMorshedy [28] | Yin [29] | Mouratidis [30] | Ram [31] | Kim [32] | Olfindo [14] | Wu [33] | Liu [34] | Wen [15] | |
| Personal properties | • | • | • | • | • | • | • | • | • | • | • |
| Density | • | • | • | • | | | | | • | • | • |
| Diversity | | • | | | • | | | | • | • | • |
| Destination accessibility | • | • | • | • | | • | • | • | • | • | • |
| Distance to transit | • | | | | • | • | • | • | | | • |
| Design | | | | | | | | • | | | • |
| Method | SEM [1] | LRM [2] | LRM | SEM | LRM | LRM | SEM | BHM [3] | ZEM [4] | LRM | MGWR [5] |

NOTE: (1) SEM: Structural equation model. (2) LRM: Linear regression model. (3) BHM: Bayesian hierarchical model. (4) ZEM: Zero equation model. (5) MGWR: Multiscale geographically weighted regression.

In order to solve these problems, this study used urban street view images and geography information to measure the built environment, and then used the MGWR model to explore the spatial heterogeneity that exists in the living environment when it affects resident satisfaction. The innovations of this paper were as follows: (1) Integrating the emerging urban street view images extraction technique with traditional geographic information technology for residential satisfaction research. (2) To apply MGWR to the study of residential satisfaction for explaining the spatial heterogeneity of environmental elements affecting residential satisfaction.

## 3. Data and Methods

### 3.1. Study Area

The present study focused on residents living in the downtown area of Zhengzhou, Henan Province. It is an important major central city in China, as well as a national integrated transportation hub, which has approximately 6.5 million people residing in its central city. This has led to excessive population density in the central city of Zhengzhou. Additionally, with rapid urban development occurring in recent years, urban development has not been coordinated; therefore, it was necessary to propose targeted improvements according to the residential satisfaction survey. The location of Zhengzhou City is presented in Figure 1. The data were collected using the random sampling method. First, a survey was conducted to collect data on residence satisfaction. Then, built environment data were obtained by extracting geographic information from the respondents' residential areas.

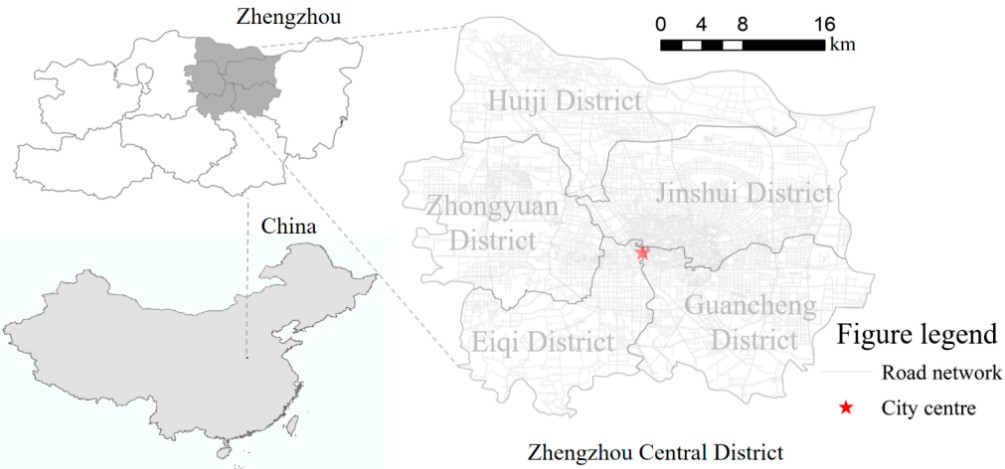

**Figure 1.** Location of Zhengzhou.

### 3.2. Research Framework

The study consisted of three steps: (I) data collection; (II) feature extraction; (III) spatial heterogeneity analysis. The study steps were as follows: Firstly, the POI, road network, and city street view data for the central city of Zhengzhou were collected through BIGEMAP and Baidu Map Open Platform. A random survey of Zhengzhou residents was then conducted to collect the results for satisfaction with living, personal property data, and residential location data. The subsequent feature extraction of the data, combined with the existing research, identified seven key indicators representing the 5D theory: bus stop accessibility [35], distance to the city center [36], distance to the metro station [3], functional diversity [4], road connectivity [37], street greenery [38], and street enclosure [39]. These indicators reflect the quality of the environment in which the respondents were living. Finally, the personal property data were incorporated with environmental indicators and the MGWR model was used to elucidate the mechanism of residential satisfaction evaluation. The reseaerch process is presented in Figure 2 and the survey questions are shown in Table 2.

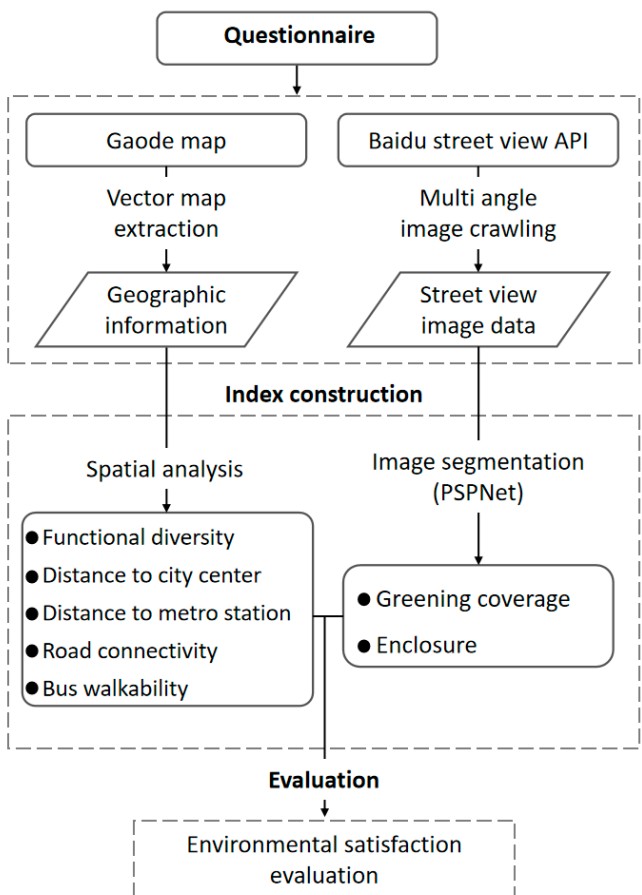

**Figure 2.** Research process.

**Table 2.** Questionnaire items for the survey.

| Questions | Options |
|---|---|
| 1. What is your gender? | Gender: male; female |
| 2. How old are you? | Age: ≤18; 18–30; 30–40; 40–55; 55–65; 65 and above |
| 3. What is your education level? | Education: primary school; middle school; high school; bachelor's degree; master's degree and above |
| 4. What is your annual income? | Income: 50,000 or less; 50,000–100,000; 100,000–150,000; 150,000–250,000; 200,000–250,000; 250,000 or more |
| 5. Where you are living? | Address: Zhengzhou City _____ District _______ Street ________ District |
| 6. Are you satisfied with the environment around your home? | Residential environment evaluation: 1; 2; 3; 4; 5; 6; 7 |

### 3.3. Variable Construction

This study examined the associations between the built environment and residential satisfaction. A widely accepted consensus for the definition of the scope of the study is the coverage of 10 min of walking time. Thus, with a walking speed of 4–5 km/h, this equated to a distance of approximately 800 m. The 800 m distance to walk has been commonly used in many studies on the urban built environment [36,40]. The maximum walkable range for this study was therefore defined as 800 m. Based on the existing studies, this study selected functional diversity, distance to the city center, road connectivity, bus stop accessibility, subway station accessibility, enclosure, and greenery as responses to environmental indicators. The built environment measurement variables are presented below (Table 3).

**Table 3.** Evaluation environment system.

| 5D Framework | Index | Description |
|---|---|---|
| Destination accessibility | Road connectivity [37,41] | The reflection of using the convenience of a road. |
| | Distance to city center [36,42] | The reflection of location in the urban area. |
| Density and diversity | Functional diversity [41,43] | The reflection of building density and the types of facilities and buildings. |
| Distance to transit | Bus stop accessibility [41,44] | The convenience of taking the bus. |
| | Subway station accessibility [3,40] | The convenience of taking the subway. |
| Design | Enclosure [38,39] | Proportion of buildings, fences, and walls in the view. The reflection of people's perceptions of space. |
| | Greenery [38,45] | Proportion of vegetation in view. The reflection of the natural environment of urban space. |

### 3.3.1. Road Connectivity

Road connectivity reflects the accessibility and relevance of urban roads within the urban space, which are studied to calculate global integration, control values, and connectivity metrics for the investigator's surrounding area with Axwoman [41]. The road connectivity within the 800 m buffer zone was then calculated using the entropy weighting method for a single road section and aggregated for the 800 m buffer zone area on a plot basis. Thereby the road connectivity was calculated for each investigator's residence [37]. The specific calculation formulae are presented in Equations (1) and (2):

$$E_i = -In(n)^{-1} \sum_{j=1}^{n} p_{ij} In\ p_{ij} \tag{1}$$

$$Q_i = \frac{E_i}{k - \sum_{i=1}^{n} E_i} \tag{2}$$

where $i$ is the index of the sample attribute; $j$ is the index of the sample; $p_{ij}$ is the proportion of attribute $j$ in the sample $i$; $n$ is the number of the samples; $E_i$ is the information entropy of the $i$th index; $k$ is the number of indicators; and $Q_i$ is the weight of the $i$th indicator.

### 3.3.2. Distance to City Center

The distance to the city center was regarded as one of the important environmental factors, which is an important factor to reflect the good or bad qualities of a residential location [36,46,47]. The measurement method was used to calculate the spatial linear distance from the neighborhood to the city center utilizing ArcGIS neighborhood analysis.

### 3.3.3. Functional Diversity

Functional diversity reflects the intensity of land development around the surveyor's neighborhood [4,48,49]. The approach was to calculate the entropy of information on eight types of POI facilities within 800 m of the community (restaurants, companies, leisure areas, schools, hospitals, government, buildings, and shopping) and to reflect the intensity of land development in the vicinity of the surveyors' residential areas. The calculation follows Equations (1) and (2).

### 3.3.4. Bus Stop Accessibility

Bus stop accessibility reflects the convenience of people choosing to travel using public transport as a significant variable in environmental measurements [50]. In China, the Planning Standards for Integrated Urban Transport Systems indicate a preference for walking distances of between 5 and 10 min, and provide the appropriate requirements for the 300 and 500 m service coverage of public transport stops. With this requirement in mind, this study considered 300 m as the optimum service radius, 300–500 m as the effective service radius, and 800 m as the maximum service radius of a bus stop. The measurement method of calculating the distance between bus stops and the surveyors'

residences is presented in Table 4, where the distance between different bus stops was aggregated and weighted to obtain the walkability of public transport.

**Table 4.** Calculation method for determining bus stop accessibility.

| Distance between Bus Stop and Residence | Walkability | Weight |
|---|---|---|
| l ≤ 300 m | $w = 1$ | 1.0 |
| 300 m < l ≤ 500 m | $w = 1 - \frac{0.5}{200}\left(l_{ij} - 300\right)$ | 0.5 ~ 1.0 |
| 500 m < l ≤ 800 m | $w = 0.5 - \frac{0.5}{300}\left(l_{ij} - 500\right)$ | 0.0 ~ 0.5 |

### 3.3.5. Subway Station Accessibility

Subway station accessibility reflects the accessibility of people who choose to travel by subway and for which it is an important environmental variable [40,51]. The measurement method was used to calculate the spatial linear distance from the neighborhood to the nearest metro station utilizing ArcGIS neighborhood analysis.

### 3.3.6. The Enclosure and Greenery

The more greenery and enclosure available, the more pleasurable and safer walking becomes. Detailed calculations were conducted based on the streets within 800 m of the investigator, with 150 m sampling intervals to generate street sampling points. Baidu Street View images were acquired from four different angles (front, back, left, and right) based on the latitude and longitude of the sampling points, and altogether 40,296 images were acquired for this study. Subsequently, extracting features were derived from street view images with machine learning algorithms utilizing the convolutional neural network tool (PSPnet [52]). The PSPnet algorithm is one of the most widely accepted semantic segmentation algorithms available to date. For the purpose of this study, GluonCV was trained on the Cityscapes dataset, which provides more suitable models for the semantic segmentation of street view images. Greenery presents the percentage of vegetation within the street view images and enclosure presents the percentage of buildings, fences, and walls within the street view images. The article utilized the surveyor's residence as the center to calculate the average of two indicators within the 800 m buffer zone in the district to reflect the evaluation concerning the surveyor's surrounding residential environment (Figure 3).

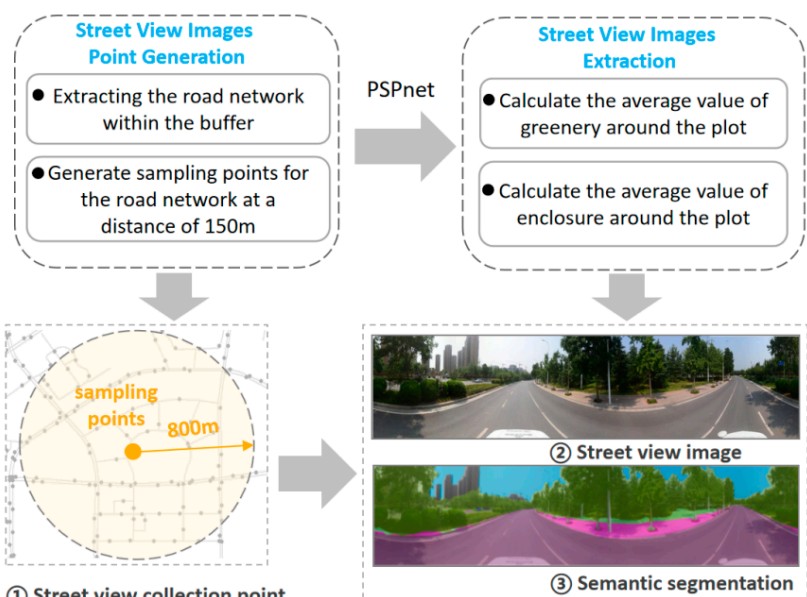

**Figure 3.** Flowchart of street view image collection and analysis.

*3.4. Spatial Regression Model*

GWR considers that spatial heterogeneity exists concerning the factors that influence resident satisfaction and explains resident satisfaction characteristics better than linear regression. In contrast, the MGWR model liberalizes the fixed bandwidth restriction compared to the GWR model, allowing further consideration of spatial scale effects; the MGWR model is calculated as shown in Equation (3):

$$y_i = \sum_{j=1}^{k} \beta_{wj}(u_i, v_i)x_{ij} + \varepsilon_i \tag{3}$$

where $y_i$ represents the weighted value of the number $i$ sample of the travel behavior latent variable; $(ug_i, v_i)$ represents the geographic marker of the number $i$ sample; $u_i$ represents the longitude of the number $i$ sample; and $v_i$ represents the latitude of the number $i$ sample. $x_{ij}$ represents the control values for the post-weighted perceived behavioral control, travel behavior intention, and built environment latent variables; $\varepsilon_i$ represents the random error; $wj$ represents the bandwidth used for the number $j$ variable regression coefficient; and $\beta_{wj}$ represents the regression coefficient of each variable.

After measuring the built environment variables around the respondents, personal attribute factors were added to analyze the quantitative relationship between the influencing factors and satisfaction with the residential environment through the application of linear regression. The MGWR model was then used to analyze the spatial differences in the coefficients of the variables, and the results were compared with the GWR and linear regression models, respectively.

## 4. Empirical Analysis

The study was conducted in December 2021 in major shopping malls in Zhengzhou and the survey was conducted using a paid approach, using cash and souvenirs as incentives. The study distributed approximately 500 surveys. After excluding the data from outside the central city of Zhengzhou and invalid samples that lacked geographic information data, 399 survey data were retained for this survey. The spatial location distribution of the respondents and the description of their attributes are presented in Figure 4 and Table 5, respectively. The built environment elements were extracted from the respondents' residential location information, and the statistical descriptions of the calculated results are presented in Table 6.

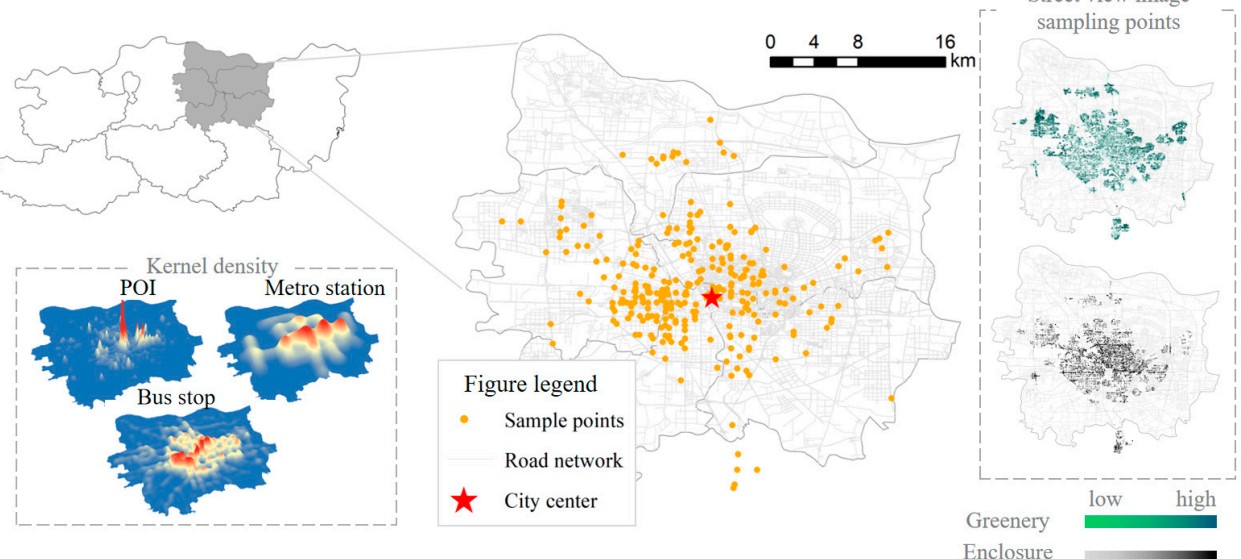

**Figure 4.** Distribution of sample locations.

**Table 5.** The statistical description of individual attribute variables.

| Statistical Variable | Classification | Number of Samples | Ratio |
|---|---|---|---|
| Gender | Male | 192 | 48.12% |
| | Female | 207 | 51.88% |
| Age | ≤18 | 10 | 2.51% |
| | 18–30 | 244 | 61.15% |
| | 30–40 | 80 | 20.05% |
| | 40–55 | 36 | 9.02% |
| | 55–65 | 12 | 3.01% |
| | 65 and above | 17 | 4.26% |
| Education | Primary school | 2 | 0.50% |
| | Junior high school | 20 | 5.01% |
| | Senior high school | 57 | 14.29% |
| | Bachelor | 275 | 68.92% |
| | Master and doctor | 45 | 11.28% |
| Personal annual income (The monetary unit is RMB. Average statistical level in 2022 is 96,400) | Within 50,000 | 182 | 45.61% |
| | 50,000–100,000 | 122 | 30.37% |
| | 100,000–150,000 | 56 | 13.83% |
| | 150,000–250,000 | 19 | 4.69% |
| | 200,000–250,000 | 7 | 1.73% |
| | 250,000 and above | 13 | 3.21% |

**Table 6.** The statistical description of environmental variables.

| Environment Variables | Average | Max | Min | SE |
|---|---|---|---|---|
| Road connectivity | 8.18 | 0.90 | 21.34 | 5.21 |
| Distance to city center | 6276.74 | 253.65 | 17,474.56 | 3923.44 |
| Functional diversity | 0.20 | 0.01 | 0.75 | 0.14 |
| Bus stop accessibility | 36.34 | 0.92 | 111.63 | 22.15 |
| Subway station accessibility | 992.55 | 10.24 | 6602.13 | 914.2 |
| Enclosure | 0.24 | 0.03 | 0.36 | 0.07 |
| Greenery | 0.16 | 0.04 | 0.35 | 0.05 |

*4.1. Validation of the Effect of Explanatory Variables Based on Linear Regression and Spatial Autocorrelation Analyses*

This study first validated the relationship between the explanatory variables and settlement satisfaction through a multiple linear regression model. Based on the results of the linear regression, the model verified the overall validity of the explanatory variables. The results show that personal attributes, such as age and education, and built environment variables, such as functional diversity, bus stop accessibility, enclosure, and greenery, all had a positive effect on residential satisfaction, while the factor of subway station accessibility had a negative effect (the further the distance from the metro station, the more the residential satisfaction was reduced). The VIF of each variable within the model appeared to be less than 10, indicating no co-linearity between the variables [50]. The Moran index was used to validate the spatial autocorrelation of the explanatory variables between different regions. According to the Moran index spatial autocorrelation test, the global Moran index values of the variables of age, education, road connectivity, distance to the city center, functional diversity, bus stop accessibility, enclosure, subway station accessibility, and greenery all presented significant spatial autocorrelation values. The explanatory variables themselves presented significant clustering in their spatial distribution and spatial characteristics that may produce differences in the perception of the quality impacting the residential environment [53]. The linear regression and Moran index calculations are presented in Table 7.

**Table 7.** Linear regression and spatial autocorrelation analyses results.

| Variables | Standard Error | Standardized Coefficient | *p*-Value | VIF | Moran I | *p*-Value of the Moran Index |
|---|---|---|---|---|---|---|
| Age | 0.05 | 0.16 | *** (3) | 1.11 | 0.17 | *** |
| Education | 0.08 | 0.13 | *** | 1.15 | 0.12 | *** |
| Personal annual income | 0.04 | −0.02 | 0.67 | 1.06 | 0.03 | 0.45 |
| Gender | 0.11 | −0.03 | 0.37 | 1.03 | −0.06 | 0.16 |
| Road connectivity | 0.01 | 0.1 | * (1) | 2.12 | 0.85 | *** |
| Distance to city center | 0 | −0.08 | 0.22 | 3.25 | 0.98 | *** |
| Functional diversity | 0.6 | 0.2 | *** | 2.5 | 0.83 | *** |
| Bus stop accessibility | 0 | 0.1 | ** (2) | 1.58 | 0.69 | *** |
| Subway station accessibility | 0 | −0.26 | *** | 1.34 | 0.73 | *** |
| Enclosure | 1.89 | 0.28 | *** | 6.5 | 0.84 | *** |
| Greenery | 1.92 | 0.41 | *** | 4.16 | 0.77 | *** |
| adj−$R^2$ | | | | 0.431 | | |
| AICc | | | | 923.04 | | |

Note: *p*-value. (1) $p \leq 0.10$: significant at 90% confidence interval (*). (2) $p \leq 0.05$: significant at 95% confidence interval (**). (3) $p \leq 0.01$: significant at 99% confidence interval (***).

### 4.2. Spatial Heterogeneity Analysis Based on MGWR

After validating the effectiveness of the linear regression model overall, as well as the spatial autocorrelation between the variables, the GWR and MGWR models were used for the spatial heterogeneity analyses, respectively. The bandwidth for each model was determined using an adaptive methodology based on Gaussian kernel functions, with the model evaluation metrics using a modified deficit pool information criterion (AICc), with lower values indicating more explanatory power for the model [25]. According to the model evaluation metrics of MGWR presented in Table 8, both $R^2$ and AICc values with the MGWR model performed better than multiple linear regression and GWR models; therefore, the MGWR model can be considered to be better than the linear regression model. The Var indicator presented in the MGWR model results was used to reflect the spatial heterogeneity of the explanatory variables, with larger Var values indicating greater variability in the spatial distribution of the effect of the corresponding variables on travel behavior.

**Table 8.** Spatial heterogeneity analysis.

| Factors | GWR | | | | | MGWR | | | | | |
|---|---|---|---|---|---|---|---|---|---|---|---|
| | Mean | Min | Max | BD (1) | VAL (2) | Mean | Min | Max | BD (1) | Var | VAL (2) |
| Age | 0.143 | 0.015 | 0.210 | 229 | 291 | 0.135 | 0.123 | 0.139 | 398 | 11% | 399 |
| Education | 0.088 | −0.004 | 0.166 | 229 | 19 | 0.051 | 0.037 | 0.063 | 398 | 41% | 0 |
| Personal annual income | 0.01 | −0.111 | 0.091 | 229 | 0 | 0.015 | 0.009 | 0.02 | 398 | 55% | 0 |
| Gender | −0.06 | −0.143 | 0.105 | 229 | 107 | −0.055 | −0.063 | −0.041 | 395 | 53% | 0 |
| Road connectivity | 0.096 | −0.212 | 0.309 | 229 | 101 | 0.118 | 0.099 | 0.129 | 391 | 23% | 327 |
| Distance to city center | −0.322 | −0.856 | 0.39 | 229 | 190 | −0.521 | −0.807 | 0.371 | 169 | 317% | 307 |
| Functional diversity | 0.145 | −0.115 | 0.649 | 229 | 137 | 0.225 | −0.389 | 0.715 | 74 | 154% | 199 |
| Bus stop accessibility | 0.09 | 0.008 | 0.187 | 229 | 21 | 0.071 | 0.056 | 0.081 | 398 | 30% | 0 |
| Enclosure | 0.179 | −0.243 | 0.645 | 229 | 47 | 0.055 | 0.034 | 0.06 | 398 | 43% | 0 |
| Subway station accessibility | −0.178 | −0.347 | −0.035 | 229 | 185 | −0.163 | −0.195 | −0.146 | 393 | 33% | 399 |
| Greenery | 0.309 | −0.005 | 0.589 | 229 | 201 | 0.191 | 0.178 | 0.212 | 398 | 16% | 375 |
| AICc | | 902.238 | | | | | 858.323 | | | | |
| adj−$R^2$ | | 0.520 | | | | | 0.542 | | | | |

(1) BD: Bandwidth, indicating spatial scale of variables. (2) VAL: Valid, the number of samples with significance at 95% confidence level. $VAL = (max − min)/|max| \cdot 100\%$.

## 5. Results and Discussion

According to Table 8, there are six factors, including age, road connectivity, distance to the city center, functional diversity, subway station accessibility, and greenery, that present strong statistical significances. The distribution of coefficients is presented in Figure 5. According to the MGWR bandwidth, the bandwidth values of the distance to the city center

and functional diversity are 169 and 74, respectively. This means that these factors are considered local variables in the MGWR model, while the other factors can be considered global variables.

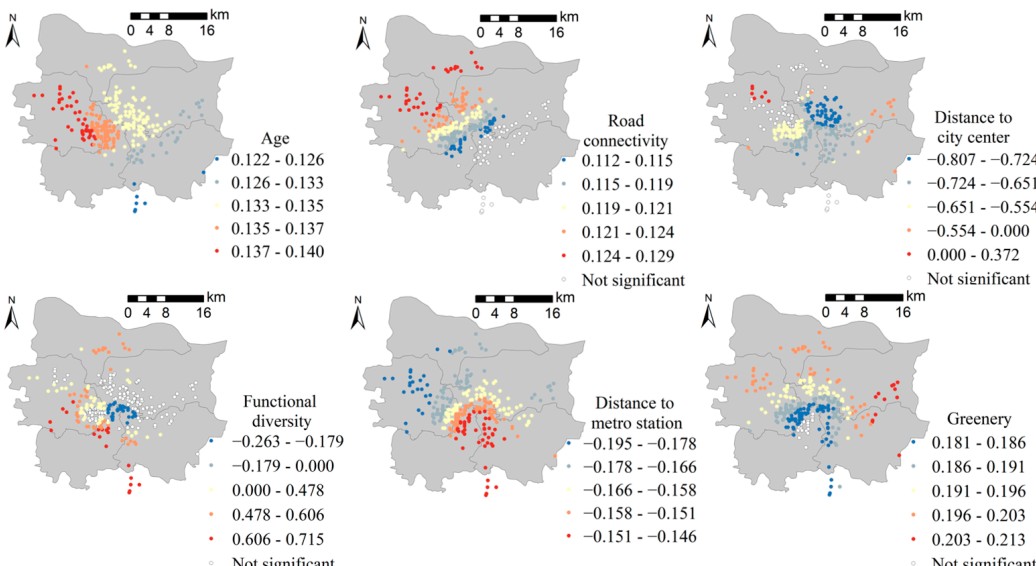

**Figure 5.** Results of spatial heterogeneity analysis.

From the perspective of building and facility distribution, the distance to the city center presents a significant negative influence in the central city of Zhengzhou (a distance further away from the city center indicates a lower satisfaction level), while the negative influence of the distance to the city center gradually decreases in the peripheral areas of the city (refer to Table 8). Functional diversity presents a negative effect on the city center (a higher functional diversity value indicates lower satisfaction levels), while there is a positive effect on the peripheral areas. This means that development occurring at excessive densities in urban areas does not increase people's residential satisfaction levels, and even presents a negative effect, which is consistent with the results of Olanrewaju [54].

It can be observed from Figure 4 that the greenery in the Zhengzhou urban centers is obviously lower than that in the surrounding areas, while the distribution of visual enclosures is contrary to this result. The factor of greenery has a significant positive effect on residential satisfaction levels, and the effect showed a consistency in space (see Figure 5). The factor of visual enclosure did not present a direct correlation with residential satisfaction. This result means that people's sense of happiness can be improved by increasing the greenery in downtown areas. Due to this result, the impact of the built environment design dimension on people's perceptions of the living environment is quantitatively confirmed.

According to Tables 7 and 8, the factors of transportation convenience present a significant correlation with residential-satisfaction levels. According to the linear regression result presented in Table 7, road connectivity is related to residential satisfaction at a 90% confidence level. The factor of bus stop accessibility did not present a significant correlation with residential satisfaction, while the factor of subway station accessibility had a significant impact on residential-satisfaction levels. This means that people living in Zhengzhou are more inclined to choose residential areas that are convenient for taking the subway, because the travel service quality of the subway is better than that of the bus in Zhengzhou. This result indicates that improving the quality of public transportation has a significant effect on improving residential satisfaction.

In personal attributes, the factor of age had significant and stable correlations with residential satisfaction in both the linear and spatial regression models (refer to Tables 7 and 8). The coefficient of age factor showed that elderly individuals are more satisfied with the residential environment than young people, which is consistent with Yang's results [55]. This result indicates that the elderly have a strong sense of belonging to a residence in

Zhengzhou, and they are easily satisfied with their living environment, while young people have higher requirements for living quality.

## 6. Conclusions

Increasing people's satisfaction with residential accommodation through improving the urban environment provides an effective instrument for enhancing people's quality of life. The study adapted classical theories and emerging technologies to explore the influence mechanism of residence satisfaction; the result can provide guidance for governments to formulate urban renewal plans to improve people's living environments.

The following points are the main contributions of this study. First, this study introduced machine vision to quantify the design dimension of a 5D framework and verified the influence of design dimension on residential satisfaction, which enriched and improved the research on built environment elements and residential satisfaction. Second, this study used MGWR to examine the spatial difference in the influence of built environment elements and verified the spatial heterogeneity in relation to residential satisfaction. This study indicated that the government needs to understand the differences in the influence of built environment elements in different regions when developing urban renewal strategies. Through the targeted identification and improvement of urban built environment elements, people's residential satisfaction levels can be effectively improved.

Based on the main results and conclusions of this study, we believe there are other areas that exist, which need to be explored in the research. First, this study only considered the two factors of greenery and visual enclosure in the design dimension of the built environment 5D framework. More diverse factors of design dimension elements should be considered in subsequent research. Second, people's socio-economic characteristics should have a significant impact on residential satisfaction; future research should further the investigation of individual socio-economic attributes and analyze the spatial heterogeneity of their impact.

**Author Contributions:** Conceptualization, Q.C. and J.C.; methodology, Y.Y. and Q.C.; software, Y.Y.; validation, Q.C.; resources, X.Z.; writing—original draft preparation, Y.Y.; writing—review and editing, Q.C. and Y.Y.; supervision, X.Z.; project administration, J.C. All authors have read and agreed to the published version of the manuscript.

**Funding:** This work was supported by the Sichuan Science and Technology Program (grant number: 2022YFH0016).

**Institutional Review Board Statement:** Not applicable.

**Informed Consent Statement:** Not applicable.

**Data Availability Statement:** Not applicable.

**Conflicts of Interest:** The authors declare no conflict of interest.

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
