# Peer review of "A Study on the Impact of Built Environment Elements on Satisfaction with Residency Whilst Considering Spatial Heterogeneity"

_sustainability, doi:10.3390/su142215011_

Round 1

Reviewer 1 Report

The paper deals with a Study on the Impact of Built Environment Elements on Satisfaction with Residency Considering Spatial Heterogeneity. The paper was a good read, however, some issues need to be solved before they can be possibly accepted.

The abstract reads and sounds causal. Even a few incomplete sentences. Please review and revise this section with a clear problem statement and your final remarks. 

The introduction is very poorly written and it is too short. Need concrete information on the topic. The information provided is very much causal. The introduction sounds like a report, please improve the style of writing. For a scientific research paper, a detailed and critical introduction is required.

Literature Review is not critically presented. Use the tabular method to show the research gaps, findings, and conclusions.

What is the Novelty of this paper? What is the major contribution concerning novelty? Further comments can be made after revision as a lot of improvement is required. Set a research methodology that details the research questions and requirements.

Results are not sufficient. Add more findings.

Separate the discussion and conclusion.- Discussion needs to be further explained (discuss the findings)

Conclusion- summaries the finding.

Scientific cohesion is missing. All more details in the paper with proper explanations. More literature is required to be addressed.  

The paper needs proofreading as many typos and grammatical errors.

Author Response

Dear professor: 
Please see the attachment

Reviewer 2 Report

Dear authors, 

Congratulations for the paper. I have some suggestions that I expect to help. 

Title: Maybe will be interesting to insert the location (China) on the title  

Introduction: I missed the goals of the paper in the introduction 

References should appear in chronological order. 

Study area: I missed a paragraph explaining the importance of the region to be studied. 

The methods are very well written. 

Maps: The maps and figures in general are too small and bright. Please improve them. 

references: DOI numbers are missing

Author Response

(The authors gave the same response as above.)

Reviewer 3 Report

The paper is interesting and relevant in the spatial analysis literature field. Its position can be recognized in the number of studies aiming to create correlations between the built environment quality and residents satisfaction. This is an interesting aspect of the actual challenges on the built environment, especially in times of pandemic: the necessity to understand how the built environment can answer to modified and enhanced necessities of people, is a strongly debated and important key topic. Given the relevance of the field of study, I don’t really find this contribution hitting and working on the core of the actual challenges on the built environment. The paper is in fact structured across the idea of analysing the correlation between the actual built environment and people satisfaction just considering the application of a very specific theory, the 5D one, that take into consideration only very few indicators and aspects both of the built environment and of people-related science (e.g. the presence of transversal services other than mobility and few others is not considered).

Moreover, I find the paper not well structured in the scientific paper format, both the subdivision of paragraphs but also its narrative form must be improved as they are hard to follow. The first parts, in particular, seems to come from different papers more than having been written following a consequent flow of argumentation.

Even if the contribution is interesting and, in general, it includes interesting elements, I found it hard to follow in some parts and to deeply understand its scope. In particular, I suggest the authors to consider the following improvements:

  • Re-writing the contribution from scratch considering a common scientific paper structure and, in particular, evidencing the gap present in the literature review, your specific research question and the method used both for the literature review and for the study in itself.
  • Deepening the scientific literature on the topic, going deeply in more actual studies on the built environment. In addition I suggest to add references to each claim authors make. Multiple times authors speculate on findings that are not well referenced (e.g. lines 30, 31, 33, and following).
  • Deepening the discussion considering that the survey has been distributed on a paid based approach, which can have partially influenced the results. In addition I suggest to discuss the fact that only one city has been considered.

Considering these points, I suggest the authors to re-write the paper, trying to make it clearer. From my point of view it has the elements for being a good contribution, but into this form I don’t think is ready for publication.

Author Response

(The authors gave the same response as above.)

Round 2

Reviewer 1 Report

Many of the previous issues have been addressed adequately. But add the abbreviation section. 

Author Response

Point 1: Many of the previous issues have been addressed adequately. But add the abbreviation section.

Response 1: Thanks to your suggestion, we checked the full abbreviation and corrected the error.

Line: 18, 137,141

Reviewer 3 Report

Dear authors, see the file for additional comments.

Author Response

Dear Reviewers
